# Acquired Hemophilia A: An Update on the Etiopathogenesis, Diagnosis, and Treatment

**DOI:** 10.3390/diagnostics13030420

**Published:** 2023-01-23

**Authors:** Ezio Zanon

**Affiliations:** Department of Medicine Via Giustiniani, Haemophilia Centre, University Hospital of Padua, 35128 Padua, Italy; ezio.zanon@unipd.it

**Keywords:** acquired hemophilia A, bypassing agents, emicizumab, COVID-19

## Abstract

Acquired haemophilia A (AHA) is a rare bleeding disorder caused by inhibitory autoantibodies against coagulation factor VIII (FVIII). AHA is a disease that most commonly affects the elderly but has also been observed in children and in the postpartum period. AHA is idiopathic in 50% of cases and is associated with autoimmune diseases, malignancies, and infections in the remaining 50%. Recently, cases of association between AHA, COVID-19 vaccination, and infection have been reported in the literature. For diagnoses, determining FVIII levels is crucial to distinguish the different causes of aPTT prolongation. Treatment of AHA is based on bypassing agents (recombinant factor VIIa, activated prothrombin complex concentrate) and porcine FVIII to control the bleeding and immunosuppressive therapy (corticosteroids, rituximab, cyclophosphamide) to suppress autoantibody production. It is important to start a prophylactic regimen to prevent further bleeding episodes until the inhibitor is negative. Recently, the series of cases reported in the literature suggest that emicizumab may provide effective and safe haemorrhage prophylaxis in the outpatient setting.

## 1. Introduction

Acquired hemophilia A (AHA) is a rare hemorrhagic disorder caused by the spontaneous development of inhibitory autoantibodies against the clotting factor VIII (FVIII) [1]. AHA has a low incidence of about 1.5 cases per million people [2,3]. AHA is a disease that most commonly affects the elderly (median age 64–78 years) but has also been observed in children and in the postpartum period [1,4,5,6,7,8,9,10]. Recently, the associations between AHA, the COVID-19 vaccination, and infection have been investigated. Mortality in AHA is estimated to be ≥20% among patients over 65 years old and those with underlying malignancies [10,11].

The clinical impact of AHA is higher because the severity of bleeding is affected by diagnostic delays and inadequate treatment [1].

Diagnoses are often delayed because the rare condition goes unnoticed by medical doctors unfamiliar with the disease [12,13]. Most patients present severe spontaneous or post-traumatic bleeding, but some are asymptomatic [14].

The management of AHA has two major goals: control bleeding by achieving adequate haemostasias and the eradication of the inhibitor [15,16]. While bleeding is being managed, efforts should be made to eradicate the inhibitor because the patient remains at significant risk of bleeding if the inhibitor is present [17]. There are many options to treat the bleeding and to eradicate the inhibitor. These options are reviewed in this paper. This review aims to update the current literature on acquired hemophilia A by reporting recent findings on possible new causes, diagnostic methods, and treatments in the acute phase and the subsequent prevention of bleeding.

## 2. Methods

Papers were found following a comprehensive search in PubMed, Google Scholar, and Scopus using the following terms for the treatment and diagnosis of the AHA, without time limits, and using the English language as a filter: “acquired haemophilia A”; “treatment”; and “diagnosis”. To collect papers about COVID-19 vaccines and hemophilia A, we used the terms “haemophilia A”, “COVID-19”, and “vaccination”. For emicizumab use, we used “Emicizumab” and “acquired haemophilia A”. The references of all retrieved original articles and reviews were assessed for additional relevant articles.

## 3. Etiopathogenesis

A total of 50% of AHA cases are idiopathic and 50% are associated with underlying medical conditions, such as autoimmune disease, malignancy. and infection, during the postpartum period as a side effect of certain drugs and vaccinations [18,19,20,21]. In the case of autoimmune disease, the development of autoantibodies against FVIII occurs mainly in rheumatoid arthritis, systemic lupus erythematosus (SLE), Sjogren’s syndrome, and dermatomyositis. No predominant oncological disease has been identified, although AHA appears to be more frequently associated with solid organ neoplasms [22]. Anti-FVIII antibodies have been described during pregnancy but are more frequent during the postpartum period up to 1 year after delivery [1,21,23]. Reported drugs associated with AHA include penicillin, phenytoin, sulfonamides, interferon, and clopidogrel [24]. In the recent literature, cases that report a possible association between AHA and COVID-19 vaccination can be found. In 2022, Cittone et al. investigated the statistical risk of diagnosing coincidental acquired hemophilia A following anti-SARS-CoV-2 vaccination [25]. The authors performed a survey within the Working Party Hemostasis of the Swiss Society of Hematology to detect cases of AHA following vaccination with the Pfizer-BioNTech SARS CoV-2 mRNA vaccine in Switzerland. The survey led to the identification of three cases of AHA. They concluded that they could not definitively state a causal relationship between anti-SARS-CoV-2 vaccines and AHA. Later, Hirsiger et al., based on the work of Cittone et al., investigated potential causal relationships between FVIII inhibition in AHA and mRNA COVID-19 vaccines. Specifically, they studied the binding, function, and cross-reactivity of the vaccine-induced anti-S-IgG in the three cases of AHA diagnosed in temporal association with COVID-19 vaccination reported previously by Cittone et al. They concluded that AHA associated with COVID mRNA vaccination was probably not due to vaccine-induced anti-S-IgG cross-reactivity [26]. Most of the studies in the literature about the relationship between AHA and mRNA COVID vaccination involved elderly patients with multiple comorbidities. All patients presented bleeding within 1–3 weeks after receiving the mRNA vaccine, either Pfizer or Moderna (mRNA-1273). Bleeding was particularly severe among patients who had completed two doses of the COVID-19 mRNA vaccine. Life-threatening bleeding, such as a large intramuscular hematoma, hemarthrosis, and even a hemothorax, were observed. Studies have also reported the appearance of AHA following COVID-19 infection [27,28,29,30]. Naturally, the confirmation of this possible association needs further evaluation.

In Table 1, we report studies about COVID-19 vaccination and AHA collected describing the symptoms, the basal value of FVIII factor activity, inhibitor titer, and treatment administrated.

## 4. Clinical Manifestations and Diagnosis

The rapid diagnosis and prompt treatment of bleeds are crucial to optimize the treatment of AHA [1,11]. AHA diagnoses are often delayed because of a lack of recognition of this rare disorder by physicians who are not familiar with the disease. Indeed, some patients are asymptomatic and only present with an isolated, prolonged activated partial thromboplastin time (aPTT) on the routine exam [42,43]. Pardos-Gea et al. reported the essential characteristics and causes of diagnostic delay in a cohort of patients with acquired hemophilia. In particular, a common issue for the delay is the use of anticoagulant/antiaggregant. In the Pardos-Gea et al. study, 28.6% had the impossibility of performing FVIII in the center of origin and the 60.7% performed the inhibitor in the center of origin. As a consequence, a statistically significant difference in diagnostic delay was found from first bleeding (<1 month vs. >1 month) regarding the mean value of days to resolve bleeding (20 (SD = 20) vs. 49 (SD = 52), *p* = 0.05) and days of hemostatic treatment (7.6 (SD = 5.7) vs. 23.8 (SD = 13), *p* = 0.003) [44]. The bleeding pattern in AHA is heterogeneous; typically, a patients’ bleeding is characterized by spontaneous hematomas, and bleeds may range from a mild condition, in which patients do not need treatment, to so severe that they are life threatening, involving limbs or organs. Approximately 30% of patients have no bleeding at diagnosis (although 94.6% in a recent European study began with hemorrhagic symptoms) [7,45,46,47]. The vast majority have clinical evidence of severe bleeding (70%) [7]. Unlike congenital hemophilia, which has a higher prevalence of joint and muscle bleeding, in AHA, the bleeding is predominantly subcutaneous [48]. The following locations are most frequently involved: subcutaneous skin and muscles of the abdomen, trunk, and limbs (large confluent hematomas) [45,46]. Gastrointestinal bleeding, post-surgical and intra-cranial bleeding, and compartment syndrome with neurological damage are the most feared events because of their high mortality rate.

Clinical manifestations do not always correlate with the level of FVIII and the titer of the inhibitor. Due to the different types of kinetics for acquired hemophilia (non-linear) compared to congenital hemophilia (linear), the inhibitor level may underestimate the inhibitor effect [47]. However, after a clinical evaluation, an isolated and prolonged aPTT with a normal PT (prothrombin time) is often the first indication of AHA, even in patients without bleeding or who have no past or family history of bleeding. The diagnosis must be confirmed by laboratory analyses. Mixing tests may be conducted if FVIII coagulant (FVIII: C) activity is not immediately available. Prolonged aPTT is often the first signal for AHA diagnoses [49]. Coagulation factor deficiencies, von Willebrand disease, lupus anticoagulation, or anticoagulation therapy may also contribute to prolonged aPTT; therefore, a differential diagnosis of AHA is imperative. Following the presentation of bleeding and a prolonged aPTT, and after excluding easily identifiable causes, such as heparin and dabigatran use, or a lupus anticoagulant (LAC)/FXII deficiency, a hematologist must be consulted. AHA FVIII inhibitors are time- and temperature-dependent, and thus aPTT results obtained immediately following the mixture of normal and patient plasma and after incubation should be compared [18]. The aPTT mixing test consists of repeating the aPTT on a sample consisting of the patient’s plasma and 50% of the laboratory’s reference plasma immediately after 2 h incubation at 37 °C. If, after the mixing test, the ratio is normalized, there will be a deficiency of a factor of the intrinsic pathway. If, on the contrary, the ratio is still abnormal, the patient most likely has pathological coagulation inhibitors. However, this test does not appear to be sufficient to rule out AHA, so further investigation must be pursued. The finding of a low FVIII: C is crucial in the diagnosis of AHA. The diagnosis is confirmed once a low FVIII inhibitor level is confirmed and established by Bethesda assay (BA), Nijmegen Bethesda assay (NBA), or in some laboratories by an enzyme-linked immunosorbent assay (ELISA) with an anti-FVIII antibody [1,19,20].

Figure 1 describes a simplified flow chart illustrating the diagnosis procedure. Alternatively, diagnoses could be conducted considering just FVIII activity. In case of FVIII activity less than 50%, it is necessary to assay anti-hFVIII (anti-human factor VII inhibitor) antibodies, the amount of the von Willebrand factor antigen (VWF: Ag), and the von Willebrand Ristocetin co-factor (VWF: Rco). If VWF: Ag and VWF: Rco are low and anti-human factor VIII is negative, the diagnosis acquired is Von Willebrand disease. If FVIII levels are equal to or greater than 50%, factor XI and IX will be assayed, and lupus anticoagulant will be distinguished [11].

## 5. Treatment

The treatment, therefore, has a three-fold objective:Stop bleeding.Prevent others by eradicating the inhibitor and the clone responsible for its production.Treat, if identified, the underlying cause.

The speed and appropriateness of the treatment and management are crucial. In the presence of hemorrhagic events, treatment aims primarily to control the acute bleeding quickly and eradicate the inhibitor, minimizing the risk of recurrent bleeding, which persists if the anti-FVIII inhibitor is present. While treating acute bleeds is a key priority in AHA care, not all patients bleed, and not all bleeds require intervention [3,11,50]. Patients with severe bleeding and a decrease in hemoglobin require immediate hemostatic treatment. Patients with mild/moderate bleeding without a significant reduction in hemoglobin may not require immediate hemostatic therapy. Still, they could develop severe bleeding at any time and thus need to be vigilant. Patients at high risk for bleeding (recent surgery, recent delivery, etc.) require prophylactic hemostatic therapy [4,16]. The introduction of bypassing agents, such as activated prothrombin complex concentrates (aPCC) (factor eight inhibitor bypassing activity (FEIBA), administered at a dose of 50–100 U/kg every 8–12 h with a maximum of 200 U/kg/24 h); recombinant activated factor VII (rFVIIa, NovoSeven, administered at a dose of 90–120 μg/kg every 2–3 h); and, more recently, FVIII porcine at an initial dose of 200 IU/kg has dramatically improved the management of acute bleeding [51].

International guidelines suggest treating the injury caused by the bleeder’s disease as quickly as possible in first-line therapy with bypassing agents, such as activated coagulation factor complicated concentrate (aPCC) or activated recombinant FVII, or with recombinant porcine FVIII [44,52,53,54,55].

### 5.1. rFVIIa

Data from the EACH2 registry also show that rFVIIa was the most widely used hemostatic agent, with an efficacy of 92% [45]. The efficacy of rFVIIa in patients with AHA in 139 patients in whom the drug was used as first-line treatment reported a median dose of 90 μg/Kg [1,56]. However, in the literature, the variability in the dose used is extensive (60–160 μg/Kg) due to the number of infusions needed (range: 1.0 to 33.0) and duration of treatment (1–7 days) [57,58]. A recent large Japanese post-marketing registry confirmed the hemostatic efficacy of rFVIIa in 92% of cases, which was statistically significantly higher when treatment started early, within a few hours of symptom onset, and in cases treated with an initial dose ≥90 μg/Kg compared with the use of lower doses [56].

### 5.2. aPCC

The effectiveness of aPCC (FEIBA^®^, Baxalta Innovations, Vienna, Austria; now Takeda) is documented by numerous case reports and a retrospective study of 34 patients in whom bleeding resolution occurred in 86% of cases [55]. In the EACH2 study, the efficacy of aPCC used as first-line treatment is similar to rVIIa (93%) [44,55]. In a recent retrospective study conducted on 56 patients by Zanon et al., aPCC, used as first-line therapy in 82% of cases at an average dose of 72.6 + 26.6 U/kg, was effective in resolving 96.4% of bleeding episodes with a median of 8 days of treatment (interquartile range [IQR] = 1–48) [59]. The combination of aPCC with antifibrinolytics has also been shown to be safe and effective in patients with cardiovascular disorders [49].

### 5.3. Human FVIII

Human FVIII concentrate can be used for hemostasis in patients with AHA, but its usefulness is limited [11,17]. It can be considered in patients with low inhibitor titers (<5 BU). It can also be used if the bypassing agents are not available immediately [4,5,18]. Some authors have proposed formulas to calculate the amount of FVIII required to overcome the inhibitors. One such recipe suggests using 40 IU/kg in addition to 20 IU/kg for each BU. If FVIII concentrates are used, plasma FVIII levels should be monitored to ensure that clinically significant levels are achieved. Calculating the amount of factor FVIII required to overcome inhibitors is of limited utility because the dosage for inhibitors can be difficult and, in some cases, inaccurate. Therefore, the extra FVIII needed is difficult to assess based on the inhibitor level [11,14,18,24,51,52,53,54].

### 5.4. Recombinant Porcine FVIII

Even when the human inhibitor is high, porcine FVIII (pFVIII) can produce detectable FVIII levels and hemostasis in AHA. Based on a prospective study in adults with serious bleeding, recombinant porcine FVIII (rpFVIII, Obizur) was approved for the treatment of bleeding in AHA in the United States, Canada, and Europe [60]. All 28 subjects showed good hemostasis and FVIII activity levels > 100% in response to rpFVIII within 24 h. A cross-reactive pFVIII inhibitor titer was required for FVIII activity in response to rpFVIII. Very high levels of FVIII activity (118–522%) were achieved by subjects who did not exhibit cross-reactivity (pFVIII inhibitor negative) [11]. The main adverse event was the development of de novo allo-inhibitors to pFVIII [45,52]. The initial dose of 200 U/kg used in the study and recommended in the product monograph may result in higher plasma FVIII levels than necessary. Subsequent doses must be adjusted according to the patient’s FVIII level response to achieve a trough level appropriate for the severity and the location of the bleeding [51,61]. Zanon et al. reported a study on a population of nine patients recruited in five Italian hemophilia centers presenting AHA and treated with intravenous susoctocog alfa (Obizur®- Takeda Pharmaceutical Company Limited. 300 Shire Way, Lexington, MA 02421 USA) as first- or second-line therapy. Safe haemostasias and rapid resolution of bleeding were observed in all cases, and the first doses used ranged from 100 to 200 IU/kg and the subsequent doses ranged between 50–100 IU/kg [62]. Additionally, Tarantino et al. reported that the hemostatic effect was achieved using loading doses of rpFVIII 100 IU/kg (6/7 patients) and subsequent doses of rpFVIII between 50–100 IU/kg, i.e., lower than recommended [63]. Ellsworth arrived at the same results in a larger case study treating 17 patients, 9 of whom received second-line therapy with an initial dose of 100 IU/kg achieving adequate control of hemostasias in all patients [64]. Susoctocog alfa was proven to be effective and safe for patients presenting severe bleeding in AHA and those with concomitant cardiovascular diseases, even if used at a lower dosage than recommended.

### 5.5. Immunosuppression

Traditional hemostasis should be achieved right away, and matter destruction should be quickly performed with corticosteroids alone or with corticosteroids and cyclophosphamide. If these counselled treatments fail or are contraindicated, patients should be treated with rituximab. Even with minor bleeding, patients are at risk for significant and fatal bleeding, regardless of initial factor VIII levels and inhibitor titers. Therefore, treatment guidelines recommend that patients receive immunosuppressive therapy immediately after diagnosis to eliminate inhibitors and normalize factor VIII levels [6,16,65]. First-line immunosuppression usually involves steroids alone or a cytotoxic drug (usually cyclophosphamide), although rituximab is increasingly used alone or in combination with other drugs [46]. “The outcome of immunosuppressive therapy depends on inhibitor eradication, risk of relapse, and adverse events including death” [46]. Inhibitors in AHA can sometimes cross-react with rpFVIII. Therefore, quantifying rpFVIII inhibitor titers should be considered before using pFVIII. Collins et al. reported the highest complete response to first-line therapy using steroids plus cyclophosphamide (66 patients from a collective of 83). Any adverse events associated with first-line treatment were higher in the patients treated with steroids alone; indeed, for patients treated with steroids plus cyclophosphamide, there were 34 events in 83 patients. In thirty-four events, four patients received intravenous cyclophosphamide, and thirty received oral cyclophosphamide [46]. To sum up, Kruse-Jarres et al. published a resume table indicating the main options for first-line immunosuppression in AHA: corticosteroids alone (prednisone 1 mg/kg orally daily or dexamethasone 40 mg orally daily 34–7 days); corticosteroids plus cyclophosphamide (cyclophosphamide 1–2 mg/kg orally daily or about 5 mg/kg 3–4 weeks); and corticosteroids plus rituximab (rituximab 375 mg/m^2^ weekly times 4, alternative 100 mg weekly times 4) [11]. They underline: “Patients with FVIII activity level <1 IU/dL at baseline require significantly longer times to remission compared to patients with FVIII activity level >1 IU/dL and may require combination immunosuppressive therapy rather than corticosteroids alone. FVIII activity and inhibitor levels should be monitored at least weekly. Applying individualized therapy according to the patient’s general condition, underlying and concomitant diseases, and prognostic factors (i.e., FVIII <1 IU/dL, inhibitor titer >20 BU/mL, presence of anti-FVIII-IgA antibodies, etc.), when available” [12]. Tiede A. et al. suggested, as an inhibitor eradication, that patients with FVIII ≥1 IU/dL and inhibitor titer ≤ 20 BU at baseline should receive first-line therapy of corticosteroids for 3–4 weeks, while patients with FVIII < 1 IU/dL or inhibitor titer > 20 BU, corticosteroids with rituximab or a cytotoxic agent have to be used as first-line therapy [49].

## 6. Bleeding Recurrences

There is therefore a risk of recurrent severe bleeding until the inhibitor is eradicated [58]. Baudo et al. reported that 25% of patients presented a second hemorrhagic event [17]. Zanon et al. showed that using aPCC in a prophylactic regimen until the inhibitor is negative prevents bleeding (3/32 in the prophylaxis group vs. 29/32 non-prophylaxis group, *p*-value = 0.0016). The mean number of days in prophylaxis after major bleeding was 12.7 (SD = 5.7) and the mean aPCC prophylaxis dose was 54.2 IU/kg (SD = 23.0) [58]. Recently, in this context, emicizumab has been proposed for the prevention of bleeding after an acute first episode and, in some cases, even in acute treatment. Emicizumab (Hemlibra^®^, also known as ACE910) is a recombinant, humanized, and bispecific monoclonal antibody that resembles the activity of FVIII. Emicizumab has a convenient pharmacokinetic profile due to its suitable subcutaneous administration route and has become interesting in the treatment of haemophilia A in patients who have or do not have inhibitors [66]. Efficacy in the prevention of bleeding was demonstrated in the HAVEN 1 study and now emicizumab is the first agent prescribed for prophylaxis in inhibitor patients who have had limited options for bleeding prevention [66,67]. The pharmacological concept of emicizumab, mimicking FVIII if its levels are low but being displaced from binding sites as soon as FVIII levels recover (due to the much higher affinity of FVIII), offers the following interesting possibilities also in AHA: prevention of spontaneous bleeding; subcutaneous application during long intervals; outpatient management of patients; and no more need for intensive immunosuppression [68,69]. In conclusion, the administration of emicizumab seems to be an important new concept for hemostatic therapy patients with AHA, with the potential to prevent bleeding, reduce side effects of immunosuppressive therapy, save costs, and protect patients from thromboembolic complications. It may also benefit nonresponding or relapsing patients instead of using third-line immunosuppressive regimens or repeated courses of treatments. Moreover, subcutaneous application of emicizumab during longer intervals offers the possibility of outpatient therapy until remission. Knoebl et al. reported on 12 patients with newly diagnosed AHA who received treatment with emicizumab [69]. The median age of the patients (6 males and 6 females) was 74 (range, 51–87) years; median initial chromogenic FVIII activity (FVIII: C; chromogenic assay using human reagents, Biophen™ FVIII: C from Hyphen) was <1%; and median maximum FVIII inhibitor titer was 22.3 (range, 3.5–2000) BU/mL. Spontaneous bleeding occurred in 11/12 patients, and surgery triggered bleeding in 6/12 patients. Emicizumab initiated a median of 8.5 (range, 2–36) days after the initial bleeding, and the median initial emicizumab dose was 2.7 (range 1.7–3.5) mg/kg. Following the loading dose, patients received emicizumab maintenance dosing of 1.5 mg/kg weekly. The bleeding stopped in a median of three (range 2–15) days in response to emicizumab; no breakthrough bleeds occurred once patients achieved FVIII: C activity >5%. Patients’ FVIII: C activity rose above 10% in a median of 11 (range 5–13) days. After a median of 105 (range 9–270) days from emicizumab initiation, FVIII: C was >50%. Indeed, in the Amandine Hansenne and Cedric Hermans case series in Belgium, according to HAVEN clinical trials, they underlined that emicizumab has a rapid efficacy in preventing bleeding, including straightforward administration and the lower possibility of aggressive immunosuppression previously associated with high morbidity and mortality and reduction in thrombotic complications [70].

The cost of emicizumab remains the single most significant barrier to its use in developing and developed countries. Multiple studies have shown that bleeding prevention with emicizumab is more cost-effective than bypassing agents [26,68]. To provide an overview of the existing literature on the efficacy of the use of emicizumab, we report in Table 2 all the cases collected on the use of emicizumab in AHA.

## 7. Conclusions

Knowledge about the recognition and treatment of acquired hemophilia A has improved in recent decades. Laboratory information allows clinicians to gather information to refine the diagnosis by eliminating confounding factors.

Currently available drugs, if used appropriately, can adequately control bleeding, even in emergencies. New pharmacological and care approaches make it possible to prevent bleeding up to inhibitor negativization and manage the patient at home. In addition, vaccination and COVID-19 infection have been reported as new causes of acquired hemophilia, but further clinical data are needed for confirmation.

## Figures and Tables

**Figure 1 diagnostics-13-00420-f001:**
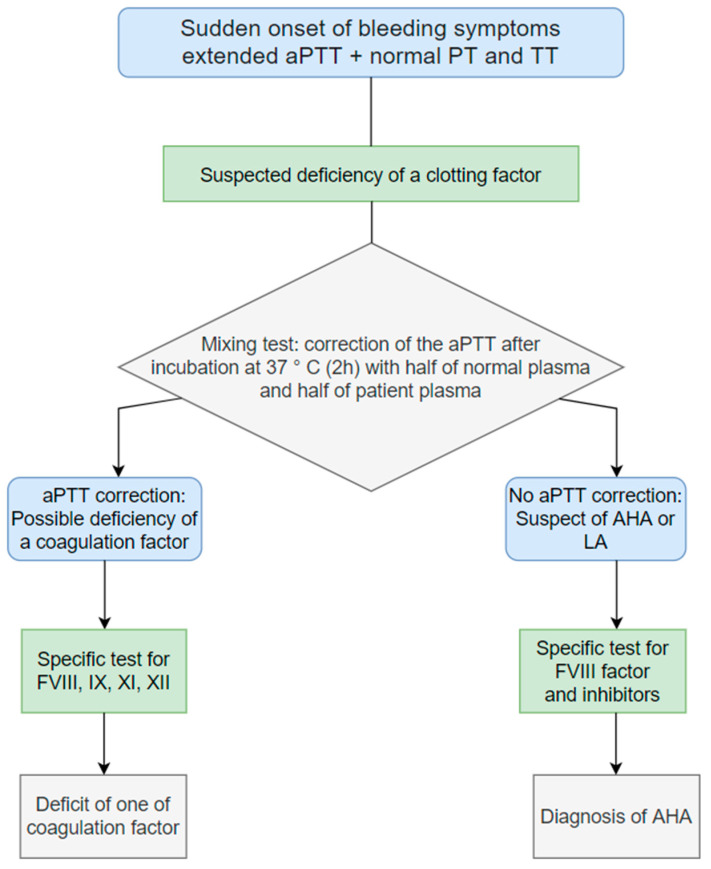
Flow chart for the diagnosis of AHA = acquired Hemophilia A. LA = lupus anticoagulant; PT = prothrombin time; TT = thrombin time; aPTT = activated partial prothrombin time; FVIII = Coagulation factor VII, IX =Factor IX, XI = Factor XI, XII = Factor XII.

**Table 1 diagnostics-13-00420-t001:** Overview of cases of acquired hemophilia A attributed to COVID-19 vaccines. aPCC = activated prothrombin complex concentrate; FEIBA = Factor VIII inhibitor bypass activity; CR = current; LA = lupus anticoagulant; PT = prothrombin time; TT = thrombin time; aPTT = activated partial thromboplastin time; rFVIla = recombinant activated factor VII; FVIII: C = FVIII coagulant activity.

Author	Age (Years) and Gender of the Patient	First/Second Dose Symptoms	Admission Laboratory Values	Treatment	Vaccine
Al Hennawi et al. [31]	75, M	Coagulopathy and bleeding into the soft tissues, distinct ecchymoses. Bleeding in the skin began three months following the second dose.	aPTT of over 90 s, PTT 114.3 s, Factor VIII activity <1%, Bethesda assay of 318 Bethesda units (BU), normal von Willebrand factor antigen level, and a 1:1 mixing study which failed to correct elevated aPTT.	rFVIla followed by prednisone 80 mg daily for 3 days, rituximab 375 mg/m^2^ cyclophosphamide 750 mg/m^2^, and cyclosporine 25 mg.	Pfizer- BioNTech SARS CoV-2 mRNA vaccine
Radwi and Farsi [32]	69, M	Mild bruising on the left wrist, 9 days after receiving the first dose. Following the second dose, several new bruises appeared on the arms and legs.	Normal PT of 10.8 s and a severely prolonged APTT at 115.2 s. The mixing study showed immediate near correction of APTT, 45 s, and prolongation of APTT upon incubation, 76.5 s. Elevated von Willebrand antigen/function, FVIII level at 1% using APTT-based assay, and FVIII inhibitor titer at 80 Bethesda units.	Prednisone (1 mg/kg) for 4 weeks followed by 5% rFVIla.	Pfizer- BioNTech SARS CoV-2 mRNA vaccine
Cittone et al. [25]	85, M	Transient pain and swelling in the right forearm 1 week after the first dose and multiple mild hematomas of the right thigh and spontaneous joint bleeding in both knees. After the second dose, the patient noted worsening hemorrhagic complications.	PT was normal, the APTT showed a significant prolongation (49s). The APTT mixing study was typical for a delayed-acting inhibitor of coagulation. Factor VIII activity (FVIII: C) was not detectable and an FVIII inhibitor was found with a titer of 2.2 BU/mL.	FFVIIa then switched to aPCC and prednisone 100 mg/day and Rituximab.	Moderna COVID-19 (mRNA—1273) vaccine
Cittone et al. [25]	86, F	Shortness of breath days after an incidental fall with chest and shoulder contusion. The patient had received the second dose 3 weeks before the fall.	Regular PT, prolongation of the APTT, with an APTT mixing study typical for a delayed-acting inhibitor of coagulation. FVIII: C was 23%, and a low-titer FVIII inhibitor of 1.01 BU/mL was detected.	rFVIla and aPCC for control of local bleeding and prednisone (1 mg/kg) FVIII:C increased to 178% after 17 days.	Moderna COVID-19 (mRNA—1273) vaccine
Cittone et al. [25]	72, F	Two weeks after having received the first dose, extensive cutaneous bruising was noticed. Ten days after the onset, the patient presented to the emergency department with multiple large cutaneous hematomas.	Prolonged APTT of 184s. Delayed-acting inhibitor in the APTT mixing study, a non-detectable FVIII activity, and a FVIII inhibitor of 12.4 BU/mL.	rFVIla, tranexamic acid, prednisone (100 mg/d), and rituximab 375 mg/m^2^ weekly (4 doses). Bleeding tendency improved 1 week after the first and third dose of rituximab. FVIII activity increased to 5%, while FVIII inhibitor decreased to 5.6 BU/mL.	Moderna COVID-19 (mRNA—1273) vaccine
Portuguese et al. [33]	76, F	Tolerated first dose. After the second dose, developed large ecchymoses covering most of her upper extremities.	Normalized APTT ratio of 1.5; the APTT was 122 s; von Willebrand factor (vWF) antigen of 5%; vWF activity <3%; and factor VIII activity <3%.	vWF/FVIII replacement therapy with Humate-P 2290 U/12 h × 4 doses IV immunoglobulin (IVIg) and methylprednisolone 125 mg. Marked elevation of vWF and factor VIII with inhibitor levels < 0.5 BU after 2 weeks of treatment.	Moderna COVID-19 (mRNA- 1273) vaccine
Farley et al. [34]	67, M	Multiple sites of cutaneous hematoma after about 20 days from the second dose.	aPTT was 72 s. A 1:1 mixing study showed no significant correction of the aPTT. Factor VIII activity was undetectable. (<1%) and an inhibitor assay confirmed the presence of a factor VIII inhibitor at 110 Bethesda Units/mL.	FEIBA at 4,500/kg/8 h, oral prednisone 90 mg, and rituximab (375 mg/m^2^/week × 4 doses). After the second dose of rituximab, FEIBA stopped at 8 Bethesda units/mL.	Pfizer- BioNTech SARS CoV-2 mRNA vaccine
Leone et al. [35]	86, M	Spontaneous disseminated hematomas with severe anemia after 14 days of the second dose.	APTT ratio 1.91; FVIII:C: 0.06 IU/mL; anti-FVIII: 2.1 Bethesda Units/mL.	Red cell transfusions and methylprednisolone therapy (1 mg/kg/day).	Pfizer- BioNTech SARS CoV-2 mRNA vaccine
Leone et al. [35]	73, F	Tongue, jaw and right knee hematomas after 26 days from the first dose.	APTT ratio 2.1; FVIII:C: 0.05 IU/mL; anti-FVIII: 0.8 Bethesda Units/mL.	Methylprednisolone therapy (1 mg/kg/die).	Pfizer- BioNTech SARS CoV-2 mRNA vaccine
Leone et al. [35]	67, M	Hematoma of the tongue extending in the cervical region 49 days from the second dose.	APTT ratio 2.55; FVIII:C: 0.06 IU/mL; anti-FVIII 2.5 Bethesda Units/mL.	Activated clotting Factor VII was administered (90 mg/kg every 6 h during active bleeding) and immunosuppressive therapy with prednisone and cyclophosphamide (both 1 mg/kg).	Pfizer- BioNTech SARS CoV-2 mRNA vaccine
Leone et al. [35]	77, M	Hematuria 52 days from the second dose.	APTT ratio 3.61; FVIII:C: 0.02 IU/mL; anti-FVIII 6.9 Bethesda Units/mL.	Activated clotting Factor VII for severe anemia (90 mg/kg every 6 h during active bleeding) and Rituximab.	Pfizer- BioNTech SARS CoV-2 mRNA vaccine
Aarya Murali et al. [36]	95, F	Spontaneous bruising over the extremities after the first dose. After 3 weeks from the second dose, presented a large hematoma on the dorsum of the right hand with resultant bleeding.	Prolonged aPTT of 83 s with normal PT of 11 s. In mixing studies, the aPTT did not fully correct and was measured at 40 s The factor VIII level was undetectable at <0.01 U/mL and the factor VIII inhibitor level was measured at 5.4 Bethesda Units/mL.	High-dose steroids (prednisone 1 mg/kg). A single dose of recombinant Factor VIII (Eloctate®- Bioverativ Therapeutics Inc. Waltham, MA 02451 USA) 2000 units. Tranexamic acid was also administered.	Vaccine: Pfizer-BioNTech SARS-CoV-2 mRNA vaccine
Fu et al. [37]	77, M	Spontaneous bruising over extremities after the first dose, hemorrhagic blisters and papules appeared on hands and trunk three weeks after receiving the second dose.	Prolonged aPTT (97.3 s), and PT (11.9 s). aPTT mixing study prolonged with incubation. The activity of the von Willebrand factor function was elevated (230%). Factor VIII (FVIII) activity lowered to 0.6%, and FVIII inhibitor titer was 71.6 Bethesda units (BU).	Prednisolone (1 mg/kg/day). Two doses of FVIII inhibitor bypassing activity (FEIBA) of recombinant factor VII activated (rFVIIa) at the dose of 90 mcg/kg were given due to persistent bleeding of the biopsy wound. Oral cyclophosphamide at a dose of 100 mg/day was added on.	Moderna COVID-19 (mRNA—1273) vaccine
Soliman et al. [38]	39, F	Lower quadrant pain and frank hematuria 10 days after the first dose. Second dose not taken.	Normal PT at 11.0 s and persistently prolonged aPTT on repeated testing ranging between 65 and 72.2 s. Lupus anticoagulant was not detected. FVII was 134.8% with persistently low factor VIII at 2%. The mixing study performed in two-time frames immediately (at zero hours) resulted in corrected aPTT followed by no correction of aPTT after 2 h incubation. The autoantibody against FVIII (FVIII inhibitor) in a titer of 17.2 Bethesda units/mL (BU/mL) was detected. Bethesda assay was repeated using the chromogenic assay and the titer was 18 BU/mL.	Prednisone (1 mg/kg) and Rituximab 375 mg/m^2^.	Pfizer- BioNTech SARS CoV-2 mRNA vaccine
Rani et al. [39]	63, M	Lower extremity swelling and pain one week after receiving the first dose, 2+ pitting edema up to the left knee, and ecchymosis to the distal mid-left thigh and posterior calf.	APTT was elevated to 68.0 s. Factor VIII activity < 1. Nijmegen assay results, 69.6	Heparin, methylprednisolone 80 mg daily and cyclophosphamide (2 mg/kg daily).	Pfizer- BioNTech SARS CoV-2 mRNA vaccine
O’Shea et al. [40]	72, M	Developed forearm, arm and thigh bruising approximately one week after receiving the first dose.	APTT of 71 s and a normal prothrombin time and fibrinogen. The lupus anticoagulant screen was negative. The Factor VIII level was reduced (0.01 IU/mL) and Factor VIII inhibitor quantification demonstrated an inhibitor of 70 BU/mL.	One dose (4500 units) of FVIII inhibitor bypassing activity (FEIBA) to control the bleeding and prednisolone 60 mg once daily [reduced dose due to age and history of diabetes]. Four weekly doses of rituximab 375 mg/m^2^ and underwent a slow steroid taper.	Pfizer- BioNTech SARS CoV-2 mRNA vaccine
Vuen et al. [41]	80, M	Presented with 4-day history of bruising over the upper and lower limbs 2 weeks after the first dose. No information is available on other doses. Second dose: not taken.	APTT of 90 s. The mixing test showed an isolated prolonged aPTT not corrected immediately or at 2 h post-incubation. Low FVIII assay of 6.7%. FVIII inhibitor assay was detectable at 7.5 Bethesda unit (BU).	Oral tranexamic acid (500 mg), methylprednisolone (500 mg daily for 3 days) and a single dose of recombinant activated FVII (rFVIIa) 90 µg/kg. Azathioprine 100 mg daily, subsequently. He was also commenced on high dose steroids (oral prednisolone 60 mg daily) in divided doses to be tapered down over six weeks. Concurrently, the patient had folate and vitamin B12 deficiency as shown in Table 1. He was given oral mecobalamin 500 µg three times a day, as intramuscular cyanocobalamin was contraindicated in this case.	Pfizer- BioNTech SARS CoV-2 mRNA vaccine

**Table 2 diagnostics-13-00420-t002:** Overview of cases of acquired haemophilia A treated with emicizumab therapy. The table reports the patients’ characteristics, admission information, dosing of emicizumab, immunosuppressive therapy (IST), clotting factor concentrate (CFC) and other treatments, the efficacy of the treatment and COVID-19-related cases. M: male; F: female; AHA: acquired haemophilia A; rpFVIII: recombinant porcine Factor VIII; rFVIIa: recombinant activated Factor VII; rhFVIIa: recombinant human factor VIIa; aPCC: activated prothrombin complex concentrate; *: Bleeding stopped; # no-adverse events; FXIII: Factor XIII.

Year	Author	Gender and Age (Years) of the Patients	Admission Information	Dosing of Emicizumab	IST	CFC and Other Treatment	Efficacy	COVID-19 Related
**2019**	Al-Banaa et al. [71]	1 F, 87	Large chest wall and pelvic hematomas	4 × 3 mg/kg weekly, 1.5 mg/kg weekly	Not reported	aPCC 50 IU/kg for 2 weeks	*#	No
	Dane et al. [72]	1 M, 72	Left anterior descending artery in-stent restenosis	Initiated at 3 mg/kg once weekly 3 days after discontinuation of FEIBA prophylaxis The patient was transitioned to emicizumab 1.5 mg/kg once weekly 28 days after emicizumab initiation. Two days after PCI, he was discharged on emicizumab 120 mg once weekly	Corticosteroids, Rituximab, cyclophosphamide, cyclosporine, azathioprine, bortezomib, mycophenolate, cladribine, and tacrolimus.	aPCC three times per week and then occasionally, rpFVIII	*	No
	Flommersfeld et al. [73]	1 F, 21	Post-surgical bleed, hematomas	4 × 3.0 mg/kg weekly, 1.5 mg/kg/wk	Dexamethasone, cyclophosphamide, ofatumumab, bortezomib, and daratumumab. ITT with IVIG and high dose FVIII substitution	rFVIIa	*#. Bleeding restarted after tooth extraction	No
	Möhnle et al. [74]	1 M, 83	Congestive heart failure and a high risk for thromboembolic and cardiac events	1 × 3 mg/kg, 2 × 1.5 mg/kg	Glucocorticoids, Rituximab	rpFVIII, rFVII aPCC, FXIII concentrate, Fibrinogen	*, Died after 36 days of emicizumab due to an arrhythmic event	No
**2020**	Escobar et al. [75]	1 M, 90 1 F, 57	Acquired Factor VIII	M: 2 × 1.5 mg/kg weekly, 1.5 mg/kg once every 21 days F: 4 × 3.0 mg/kg weekly, 1.5 mg/kg weekly	Not reported	Not reported	*	No
	Hess et al. [76]	1 M, 91	Ongoing hematuria for 5 weeks with prior workup unrevealing.	4× 3 mg/kg weekly, 2 × 1.5 mg/kg weekly	Prednisone, Cyclosporine.	rFVIIa 90 mcg/kg every 2 h for a total duration of 24 h	*#	No
**2021**	Al-Banaa et al. [77]	1 M, 79	Symptomatic anemia associated with bleeding is thought to be anticoagulation-related.	4 × 3.0 mg/kg weekly, 3 μg/kg every 2 weeks	Prednisone.	Total of four doses (100–200 U/kg) of rpFVIII	*, Deep venous thrombosis after several weeks of emicizumab maintenance therapy	No
	Chen et al. [78]	1 F, 57 1 M, 67 1 M, 74 1 M, 68	1 F: Thigh hematoma, anemia of blood loss 1 M: Skin hematomas (abdomen), gastrointestinal bleeding, epistaxis 1 M: Hemarthrosis of the right knee, skin hematomas 1 M: Skin hematomas.	4 × 3.0 mg/kg weekly, 1.5 mg/kg weekly	Rituximab (all patients) Prednisone (1 M patient) Cyclophosphamide (1 M patient and 1 F patient).	1 F:23 doses(rFVIIa) and 27 doses (rpFVIII) 1 M:48 doses (rpFVIII) 1 M:25 doses (rpFVIII) 1 M:12 doses (rpFVIII)	*#	No
	Ganslmeier et al. [79]	1 M, 59	Diarrhea and abdominal pain.	Every four weeks 300 mg	Cyclophosphamide (1000 mg per cycle) with concomitant prednisolone therapy, corticosteroids (starting with 250 mg tapered down to 30 mg per 24 h), two cycles of Rituximab, tranexamic acid.	rFVIIa	*	No
	Hansenne & Hermans [70]	1 M, 73 (A) 1 M, 93 (B)	A: Multiple skin hematomas, along with a large muscle and soft tissue hematoma of his left thigh. B: Acute onset hemorrhagic diathesis.	A: 13.0 mg/kg weekly, 6.0 mg/kg once B: 4 × 3.0 mg/kg weekly, 2 × 3.0 mg/kg weekly (4 doses)	A: Rituximab (375 mg/m^2^, weekly, four doses in total) and cyclosporine (100 mg per day for 1 month), treatment with 1 mg/kg methylprednisolone was initiated on day 5 following his admission. B: methylprednisolone (from day 1) as immunosuppressive treatment and 375-mg/m2 Rituximab (from day 2, weekly for four doses).	A: 7 mg of rhFVIIa B: No additional hemostatic agents were required	*	No
	Jena et al. [80]	1 M, 78	Intrahepatic biliary radical dilatation on ultrasound during a routine health checkup.	Not reported	Steroids, cyclophosphamide.	rFVIIa, aPCC	*	No
	Knöbl et al. [67]	6 M, 6 F The median age was 74 years (range 51–87).	Newly diagnosed AHA. 8 patients showed Severe bleeding, 6 bleeding associated with surgical wounds.	3.0 mg/kg weekly (2–3 doses), 1.5 mg/kg/3 weeks	Steroids, Rituximab, cyclophosphamide.	rFVIIa	*, stroke in 1 patient during emicizumab	No
**2022**	Chen et al. [81]	5 M, 6 F, median age was 77 (range 47–93)	AHA, 8 patients experienced bleeding at >1 site.	4 × 3 mg/kg of emicizumab, except for one that continued emicizumab every two weeks to complete three months of treatment per clinician discretion; one had insurance approval for only two doses of emicizumab	All patients received four weekly doses of 375 mg/m^2^ of Rituximab.	6 doses of rFVIIa (On or before starting emicizumab)	*#. One experienced rebleeding	Yes
	Crossette-Thambiah et al. [82]	A cluster of three AHA patients	AHA.	Not reported	Each patient received a BNT162b2 (Pfizer) vaccination. Bypassing therapy and steroids.	rpFVIII	*	Yes
	Happaerts & Vanassche [83]	1 M, 75	Multiple hematomas, hemorrhagic bullous pemphigoid, and a gastrointestinal ulcer.	2 × 3.0 mg/kg	Methylprednisolone 64 mg daily, Rituximab 375 mg/m^2^ weekly (2 doses) Sars-Cov2 vaccination (AstraZeneca).	rFVII	*, new AF, acute kidney injury, and methicillin-sensitive Staphylococcus aureus sepsis and the end dead	Yes
	Knöbl et al. [84]	11 M, 9 F, median age 79 (range 51–87)	AHA.	4 × 3 mg/kg weekly and 1.5 mg/kg 2–4 weeks intervals	Steroids, Rituximab.	rhFVIIa	*#	No
	Latef et al. [85]	1 F, middle-age	Human immunodeficiency virus (HIV) developed refractory hemophilia with bleeding episodes.	4 × 3.0 mg/kg weekly,1.5 mg/kg/wk	Corticosteroids, cyclophosphamide.	Bypassing agents	*#	No
	Shima et al. [86]	12 patients	AHA.	6 mg/kg (day 1), 3 mg/kg (day 2), 1.5 mg/kg weekly (from day 8 onwards)	Not reported.	Not reported	For 10/12 patients *#. 5 minor bleeds in 2 patients	No
	Yates et al. [87]	1 M, 83	Fatigue and weakness, which were attributed to anemia	Not specified	Prednisone (70 mg, daily), cyclophosphamide (75 mg, daily), Rituximab (375 mg/m^2^, every week for 4-weeks).	rFVIIa; 90 mcg/kg every 6 h	*#	No

## Data Availability

Not applicable.

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
