# Peer review of "Acquired Hemophilia A: An Update on the Etiopathogenesis, Diagnosis, and Treatment"

_diagnostics, 2023, doi:10.3390/diagnostics13030420_

Round 1
Reviewer 1 Report
The author gives in this review article a overview on the aetiology, diagnosis and management of acquired haemophilia. A rare clotting disease. Where recognition of this disease is often difficult. The possible relation between aqcuired haemophilia and COVID19 vaccination is addressed. Furtermore possibities for vtreatment of bleeds, including reported emicizumab. Furthermore the overall treatment of the inhibitor, where a shift from cyclophosphamide to Rituximab or MMF is discussed.
It is an overview of the current and possible future treament.
Some suggestions:
line 43: reference 13 seems not correct
line 49/50: the percentages should be reversed.
line 55: Suggestion to compare it to congenital haemophilia: joint bleeding , muscle bleeding and much less subcutaneous bleeds.
line 64/65: I could not find the statement of a correlation of FVIII and bleeding. I assume that the message found in the reference can be changed to "Due to the different type of kinetics for aquired haemophilia (non lineair) compared to congential hemophilia (linear), inhibitor level can underestimate the inhibitor effect.
line 73/74. I would suggest that in case of a presentation of bleeding and a prolonged apTT, after excluding an easily identifcable causes like heparin, Dabiagtran use, or a LAC / FXII deficiency, an hematologist must be consulted. As most diagnoses will be treated by the hematologist.
line 112/113: As it is formulated here it seems that the authors earlier found an assiociation between vaccination and AH (but it was not due to anti-S-IgG). however in their previous article (https://doi.org/10.1111/jth.15421) they think there is no causal relation. (However they could not definitively rule out a causative relationship.)
line 152: as far as I know dosing of FEIBA is mostly listed as 50-100 IU /kg.
line 193- 196: Is difficult to understand. As I read it: the cause and effect are reversed. I assume first bleeding risk cannot be predicted based on FVIII/ inhibitor level. furthermore inhibitor levels are difficult / inaccurate. Therefore the needed extra FVIII needed is difficult te assess based on inhibitor level.
line 240: I would ad: Often cross reacting FVIII inhibitors against pFVIII are found. Therefore quantification of rpFVIII inhibitor titres should be considered before use of pFVIII.
line 225: Tiede et al in 2020 guideline did not make a difference for the first line treatment between rituximab or cyclofosfamide. I have the impression that nowadays most treaters use the combination prednisolon / rituximab instead of prednisolon / cyclofosfamide. I would suggest to use the suggestion of Tiede and give both options.
Author Response
In the uploaded Word file, you can find the answers to each point raised by Reviewer 1.

Reviewer 2 Report
The author has undertaken a literature review on acquired haemophilia A. There are several reviews that have previously been undertaken on this topic including one by an international panel of AHA experts in 2017. This publication provided a comprehensive diagnostic pathway. I don't see the merit in providing a partially complete pathway in the submitted manuscript, and it is remiss not to have referenced the international panel's paper in association with the diagnostic pathway section.
The author has added to the literature by undertaking a review of the association of AHA and COVID-19 vaccination. Although cases have been published of AHA occurring with COVID-19 infection, this topic received an extremely brief mention without references provided. I don't know why this didn't receive more attention.
The paper would require substantial additional work on the structure, the quality of writing and referencing before being acceptable for publication. I have not provided suggested edits for the complete manuscript as it would take too much time, however, below provides examples of the improvements that need to be made:
The first 8 lines of the manuscript (24-31) read as a list of disconnected brief statements and should be rewritten to flow and interconnect in a logical manner.
Line 25 - introduce the autoantibodies as 'inhibitory autoantibodies, known as inhibitors' so there is a connection to the subsequent term 'inhibitor'.
The description (below) of the search methods is rather disruptive in the Introduction section and should appear in a Methods section. Additionally, the term 'we' is used, however only one author is listed and there are no acknowledgements.
We conducted a comprehensive search in PubMed, Google Scholar and Scopus using the 34 following terms for the treatment and diagnosis of the AHA, without time limits, and 35 using the English language as a filter: “acquired haemophilia A” AND “treatment” AND 36 “diagnosis”. To collect papers about COVID-19 vaccines and haemophilia A, we used the terms “haemophilia A” AND “COVID-19” AND “vaccination”. For emicizumab use, we used “emicizumab” AND “acquired haemophilia A”. The references of all retrieved original articles and reviews were assessed for additional relevant articles.
The Introduction section should contain additional summary information such as what appears in the abstract:
'AHA is a disease that most commonly affects the elderly but has also been studied observed in children and in the postpartum period. AHA is idiopathic in 50% per cent of cases and otherwise is associated with autoimmune diseases, malignancies, and infections in the remaining 50% per cent. Recently, cases of association between AHA and COVID-19 vaccination and infection have been reported in the literature.'
The corrections appearing in the paragraph above should be made to the abstract.
The Etiopathogenesis section should come after the Introduction section and prior to the section on clinical manifestations and diagnosis. This suggested order also aligns with the order in the title.
The opening sentence in the Etiopathogenesis section should be rewritten as it is not convention to start a section or sentence with a numeral.
The wording could also be improved upon throughout the manuscript e.g. '...AHA cases are associated with underlying medical conditions, such as autoimmune disease, malignancy, and infections, during the postpartum period, and as a side-effect of certain drugs and vaccination...'
and
'In the case of autoimmune disease, the development of autoantibodies against FVIII occurs mainly has been observed in rheumatoid arthritis, Systemic lupus erythematosus (SLE), Sjogren’s syndrome and dermatomyositis. There is n No predominant oncological disease has been identified, although it seems more AHA appears more frequently in association with solid organ neoplasms [25]. Drugs reported associated with AHA has been reported following the use of included penicillin, phenytoin, sulfonamides, interferon, and clopidogrel...
The order of the topics in the first paragraph of the Etiopathology section need to be aligned with the order in which they are presented in the opening sentence. Have the drugs and COVID-19 vaccination sentences following each other, not split by the pregnancy and postpartum sentence. Provide a reference(s) for the pregnancy and postpartum period statement.
Line 111: 'In recent literature, we can find cases that report the possible association between AHA and COVID-19 vaccination.'
Where are the references for these cases?
Who is 'we' - co-authors/acknowledgements?
Lines 112 and 113: 'In April 2022, Hirsiger et al. investigated potential causal relationships between FVIII inhibition in acquired haemophilia A with mRNA COVID-19 vaccines[26].' A brief description of the methods used should be provided for this study.
The following statement appears on lines 114 and 115 in the submitted manuscript:
'They concluded that AHA is associated with mRNA COVID vaccination was likely not due to vaccine-induced cross-reactive, FVIII inhibiting anti-S-IgG.'
The following is the conclusion in the publication you refer to:
'We conclude that AHA associated with mRNA COVID vaccination was likely not due to vaccine‐induced cross‐reactive, FVIII‐inhibiting anti‐S‐IgG.'
It is appropriate to describe the conclusions in your own words and in greater detail as, in the absence of the accompanying text that appears in the publication, this sentence is inadequate for the comprehension of the reader.
Lines 116-122: Where is the reference? Reference 26, the only one that has been provided for the text on the association of COVID-19 vaccination and AHA, included just 3 patients.
Line 124: FVII or FVIII? factor activity or coagulant activity? change 'administrate' to 'administered'.
The following paragraph needs rewriting as it consists of two sentences and is impossible to comprehend.
'Figure 1 describes a simplified flow chart illustrating the diagnosis procedure. Alternatively, the diagnosis could be conducted considering just the FVIII activity less than 50%, in this case, it is possible to assay anti-hFVIII (anti-human factor VII inhibitor) antibodies, the amount of the von Willebrand factor antigen (VWF: Ag) and von Willebrand factor-Ristocetin co-factor (VWF: Rco) taking into account a low VWF: Rco to diagnose acquired Von Willebrand disease (VWD), thus distinguishing the diagnosis from that of AHA. If FVIII levels are equal to or greater than 50%, factor XI and IX will be assayed and lupus anticoagulant will be distinguished[12].'
As mentioned above, this is just an example of the short-comings of this paper. The whole manuscript needs to be substantially improved in all of the areas outlined.
Author Response
In the uploaded word file, you can find the answers to each point raised by Reviewer 2.
